# The Role of Trees in Winter Air Purification on Children's Routes to School

**Adrian Hoppa** [1], **Daria Sikorska** [1,*], **Arkadiusz Przybysz** [2], **Marta Melon** [1] and **Piotr Sikorski** [1]

1 Department of Remote Sensing and Environmental Assessment, Institute of Environmental Engineering, Warsaw University of Life Sciences-SGGW, Nowoursynowska 159 Str., 02-776 Warsaw, Poland; adrian_hoppa@sggw.edu.pl (A.H.); marta_melon@sggw.edu.pl (M.M.); piotr_sikorski@sggw.edu.pl (P.S.)

2 Institute of Horticultural Sciences, Warsaw University of Life Sciences-SGGW, Nowoursynowska 159, 02-776 Warsaw, Poland; arkadiusz_przybysz@sggw.edu.pl

* Correspondence: daria_sikorska@sggw.edu.pl; Tel.: +48-2259-35393

**Abstract:** Air pollution is now considered to be the world's largest environmental health threat, accounting for millions of deaths globally each year. The social group that is particularly exposed to the harmful effects of air pollution is children. Their vulnerability results from higher breathing frequency and being subject to concentration peaks just above the ground. The negative effects of ambient particulate matter also depend on the time of exposure. A daily route to school can constitute an important component of children's physical activity, but air pollution can pose a threat to their health. Numerous studies have proved that high loads of PM can be effectively reduced by vegetation. Little is known, however, on whether vegetation can also reduce PM during leaf dormancy. In this study we investigated the role of trees in air purification during the leafless period in children's routes to selected schools located in Warsaw during winter. The results obtained show a weak impact of the tree canopy in winter.

**Keywords:** air pollution; particulate matter; WAI; urban trees; brown leaf area index

## 1. Introduction

Air pollution is now considered to be the world's largest environmental health threat, accounting for millions of deaths globally each year [1–3]. The main component of air pollution is particulate matter (PM), which can be emitted to the atmosphere directly (primary PM), or can be formed as a result of chemical reactions (secondary PM) [4]. Apart from natural sources of PM, the anthropogenic PM emissions include primarily fuel combustion and manufacturing processes [5,6]. Recent years have shown that the highest annual average concentrations of $PM_{10}$ and $PM_{2.5}$ in Europe occur in central and eastern European countries, mainly in Poland [7].

Among various air pollutants, particulate matter (PM), because of their small particle size, is the most harmful and most representative pollutant [8,9] and its major toxicological effects on human health and the environment have been observed for decades [10]. PM has been associated with an increased risk of respiratory health outcomes among children [11,12] and an increased risk of cardiovascular diseases, including heart failure and myocardial infarction, hypertension and stroke [13]. Children are a particularly vulnerable group to the effects of PM [14], as they are more active, breathe proportionately more air than adults, their respiratory systems are still developing, and they spend more time outdoors, inhaling the highest PM concentrations just above the ground. Children growing up in the most polluted areas reveal significant lung function deficits [15] and studies show an increased incidence of ADHD [16] and developing allergies [17]. The locations particularly important in terms of risks of children's exposure to air pollution are their routes to school. Walking to school takes up more than 50% of children's active time, compared to about 20% spent at school, 10% at home and 1% in green areas [18].

Concerns over the health and well-being of the city residents, particularly the most vulnerable groups, make it necessary to take appropriate measures to shield them from exposure to harmful PM. As a significant share of the children's daily activities is their commute to school, the proper design of their routes can support a friendly and healthy environment and reduce the negative effects of air pollution on their journey [19]. The selection of proper plant species in these areas can have a positive effect on the ambient air quality, particularly where PM levels are exceeded [20]. Positive relations between the presence of greenery and their beneficial role in ensuring children's health and well-being have been long investigated. Children who live near urban green areas have better lung capacity [21], while street trees have been proven to be beneficial for childhood asthma prevention [22]. Growing up in greener neighborhoods may also be beneficial for brain development and cognitive functions [23]. Children who grew up in environments with the lowest levels of green were 55% more likely to develop mental disorders [24].

Factors that influence the ability of plants to accumulate pollutants are the location and structure of greenery, morphological characteristics of plants forming the plant community and environmental conditions [25]. Plants do not have the ability to escape from a contaminated site, and therefore they have evolved mechanisms that allow them to survive in a contaminated environment. This ability is the basis of phytoremediation technology, which involves using plants to trap pollutants and, under certain conditions, break them down. Phytoremediation uses selected species of trees, shrubs and climbers that are able to accumulate on their leaves PM harmful to human health, thus supporting the process of air purification from pollutants [26]. Research shows that the presence of plants near buildings can have a positive impact on well-being, as well as physical and mental health [27]. However, during the winter, when air emissions are particularly high [28], there is a lack of comprehensive research on the role of trees in the leafless season, which makes the role of plants in the winter season unclear. The aim of this study was to determine the extent of particulate matter pollution that children are exposed to on their routes to school during the leafless season, and to investigate how the adjacent trees can reduce exposure to high PM concentrations.

## 2. Materials and Methods

The study area is located in Warsaw, Poland's capital and the largest city with a population of 1.79 million [29]. The average annual temperature in Warsaw is about 9.3 °C, while the yearly precipitation is about 695 mm. Despite being a relatively green city, with the vegetation cover exceeding 50% [30], Warsaw is characterized by the phenomenon of urban heat island [31]. The character of pollution in Warsaw is typical of large urban agglomerations, $PM_{2.5}$, $PM_{10}$, nitrogen and carbon oxides and sulfur dioxide being the dominant pollutants [32]. In the very center of Warsaw and in densely populated districts located outside the city center, high levels of PM have been noted. The analysis of PM concentrations in the winter season showed that the main emission sources are of anthropogenic origin (energy production based on coal and biomass combustion). In the warm season, the pollutants mainly originate from local emission sources, mainly urban traffic and transportation [33]. Warsaw launched the program to improve air quality and has been continuously encouraging and subsidizing the removal of black-smoke-belching stoves; however, the share of existing ones still remains at a high level, being unevenly distributed throughout the city (Figure 1).

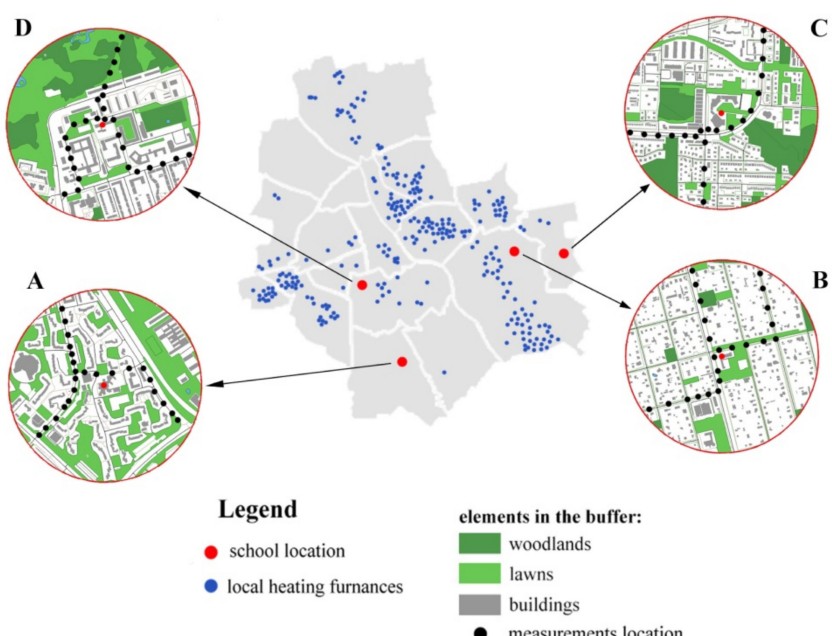

**Figure 1.** Location of the selected schools and the examined routes and study plots. Examined schools (**A**)—no. 319 (Ursynów district), (**B**)—no. 218 (Wawer district), (**C**)—no. 385 (Wesoła district), (**D**)—no. 70 (Mokotów district).

In this study we investigated the pollution levels during a daily school commute of children on their way to school. We took into consideration primary schools, due to the fact that children aged 6–14 are more susceptible to high pollutant concentration levels, but they are also most likely to commute to school on foot. Due to regionalization of primary education in Poland, those children are more likely to attend the nearest school. Out of 320 elementary schools in Warsaw [34], we selected four that were located outside the strict city center, so that the mean annual pollution levels did not differ significantly (Table 1). We selected schools representing various possible pollution levels originating from municipal emissions and similar conditions in terms of possible traffic (proximity to larger roads). The schools were selected in pairs—one group near a cluster of black-smoke-belching stoves (B and D) and one beyond such a cluster (A and C) (Table 1). One pair (B and C) was selected in a location where the share of tree-covered areas in the neighbourhood (500 m buffer zone) was high (over 30%) and the other schools (A and D) were characterized by a lower share of tree-covered areas in their immediate vicinity (Table 1). For the assessment of the share of vegetated surfaces and traffic conditions, we used BDOT (Database of Topographic Objects for Poland), which is the most fundamental source of information on the location of topographic objects in Poland [35].

**Table 1.** Characteristics of investigated school locations and neighborhood in a 500 m buffer zone (source: BDOT 10k) and average annual concentrations of $PM_{2.5}$ and $PM_{10}$ [36] and locations of black-smoke-belching stoves—heating furnaces [37].

| School | District | Share of Tree Covered Area in a 500 m Buffer Zone(%) | Mean Annual Concentration of $PM_{2.5}$ ($\mu g/m^3$) | Mean Annual Concentration of $PM_{10}$ ($\mu g/m^3$) | Number of Black-Smoke-Belching Stoves in 1 Km Buffer |
|---|---|---|---|---|---|
| A | Ursynów | 21.1 | 20.5 | 26.8 | 0 |
| B | Wawer | 36.6 | 19.6 | 25.4 | 8 |
| C | Wesoła | 57.2 | 19.1 | 24.5 | 0 |
| D | Mokotów | 0.7 | 22.1 | 29.0 | 7 |

We inventoried all walking routes from the schools' entrance within a buffer of 400 m that we observed to be frequently used by children as their home-school routes (after initial observations conducted during one workday at each of the selected schools). In each of the locations, we chose 3 to 4 of the main routes most frequently used by the pupils.

Along the routes, we took regular measurements in plots located every 20 m. In each plot we identified winter vegetation density (LAI) and measured PM concentrations. In our research we took into account the locations under the tree canopy and beyond the tree cover. In cases when there was another walking route crossing the path, or where the plot was located at the edge of the tree canopy, the plot was rejected to avoid the edge effect.

At all locations we measured $PM_{2.5}$ and $PM_{10}$ concentrations in each of the study plots in December 2020 on three windless days (nearest local weather station indicating winds below 0.2 m/s) at weekly intervals. Measurements for each day were made in the morning (8–10:00), representing highest concentration rates typical for peak hours (possible increased loads due to traffic) and in the afternoon (12–14 PM) hours at the time when children were travelling from school, but also when the PM concentrations recorded could be lower and originate further from the heating sources.

We measured $PM_{2.5}$ and $PM_{10}$ concentrations with the Dust Air device [38] at 140 cm, corresponding to the height of the primary school pupils. For each school we recorded a series of measurements in 3 days, every week, meaning there were 3 repetitions per each school, concerning both morning and afternoon measurements. Each series lasted 60 s with a 10 s interval between the measurements. In order to determine the relation between vegetation density and PM loads we measured Leaf Area Index (LAI) along the selected home-school routes using the SS1-COM-R4 Complete System with Radio Link [39]. The measurements were taken along the walking routes on their right side in the direction of walking, the study plots were located 1.5 m away from the side of the pavement. As the LAI measurement is being calculated per 1 $m^2$, to ensure we captured the diversity of vegetation density we took 3 adjacent measurements and averaged the score per plot. The LAI meter is most commonly used to determine the density of canopy; however, during the vegetation dormancy season it can be effectively used for the assessment of the density of branches, which could allow for the deposition of PM and also a sheltering of the walking routes from high loads of pollutants. LAI measured during leafless season is referred to as Wooden Area Index (WAI), which provides information on the density of woody shoots and leaves remaining in the winter [40].

*Statistical Analysis*

We analysed the data on PM concentration and vegetation density in Statistica 10 software. We tested the relationship between WAI and PM concentration levels with Pearson's correlation (after having confirmed the data is normally distributed with Kolmogorov–Smirnov test). We used one-way ANOVA for comparing the data at $p < 0.05$ significance level.

## 3. Results

### 3.1. Particulate Matter Content of the Studied Routes to Schools

Ambient air PM concentrations recorded along routes to school in wintertime were very much associated with the location of the school in terms of the proximity to individual household heating emitters (Table 1). Irrespective of the PM fraction and the time of day when measurements were made, significantly higher PM concentrations were recorded at schools B and D, which were surrounded by more emission sources (Figure 2). Concentrations at these locations exceeded the average acceptable level for the 24-h period ($PM_{10}$ 50 $\mu g/m^3$). Concentrations recorded in the afternoon were significantly higher than in the morning during rush hours (Figure 2).

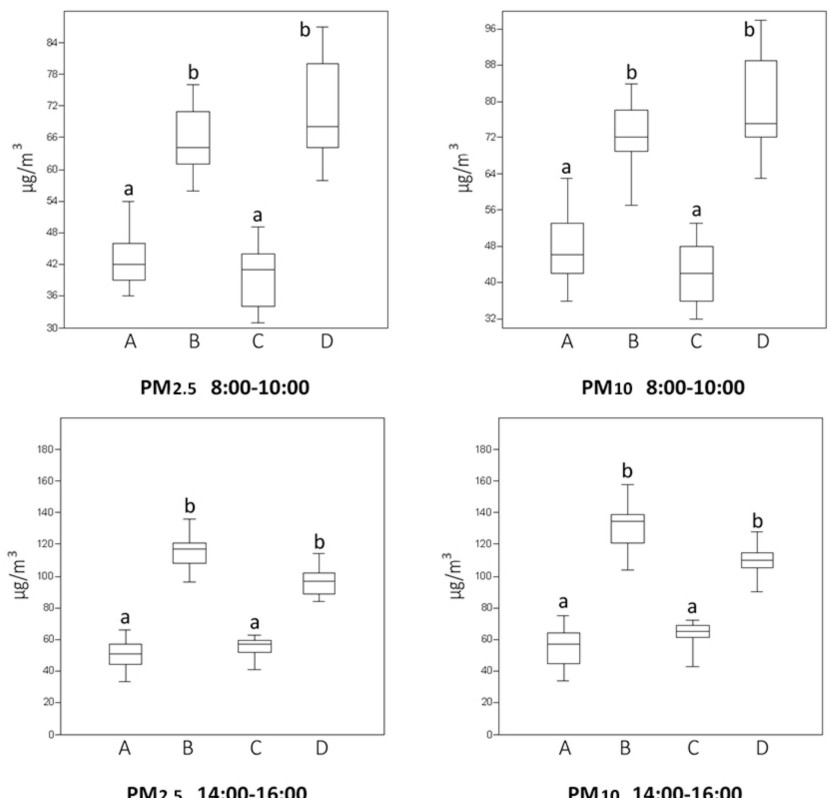

**Figure 2.** Mean PM$_{2.5}$ and PM$_{10}$ ($\mu$g/m$^3$) concentrations of in the morning and evening on the studied sections of the road to schools from all measurement points. a, b—homogeneous groups, significant differences at $p < 0.05$. Letters A–D refer to school's symbols.

*3.2. Vegetation Density along the School Routes in Winter*

There were no statistical differences in the amount of WAI along the studied routes to schools. The average index ranged from 0.2 to 0.6, with high spatial variability (Figure 3). The highest mean WAI values were recorded in the surroundings of schools A and C (Figure 3), where the proportion of forests and woodlands was the highest (Table 1).

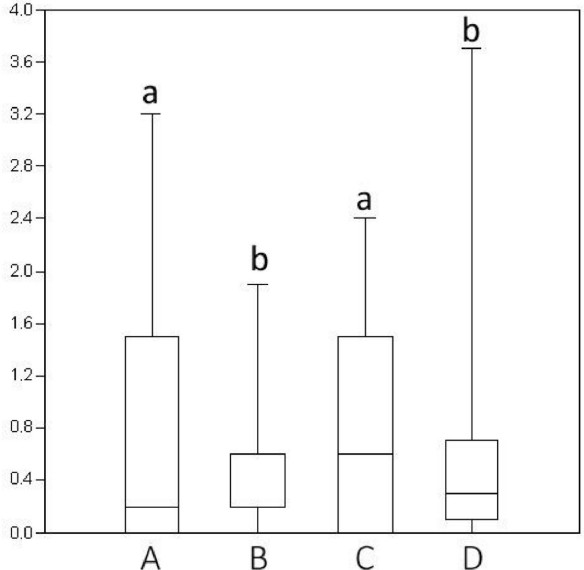

**Figure 3.** Comparison of the WAI along the studied routes to schools. a, b—homogeneous groups at $p < 0.05$.

### 3.3. Relation between PM Concentration and Vegetation Density in Winter

We found an ambiguous negative relationship between the greenery and PM concentrations during the vegetation dormancy period (Figures 4–7). On one hand we found a significant effect (Figures 4–7) in the morning hours for schools A and D (Figures 4 and 7), in sites with a low proportion of trees in their neighbourhood (Table 1). The relationship between WAI and PM concentration was positive, WAI related to a local increase in both $PM_{2.5}$ and $PM_{10}$ pollution (Figures 4 and 7).

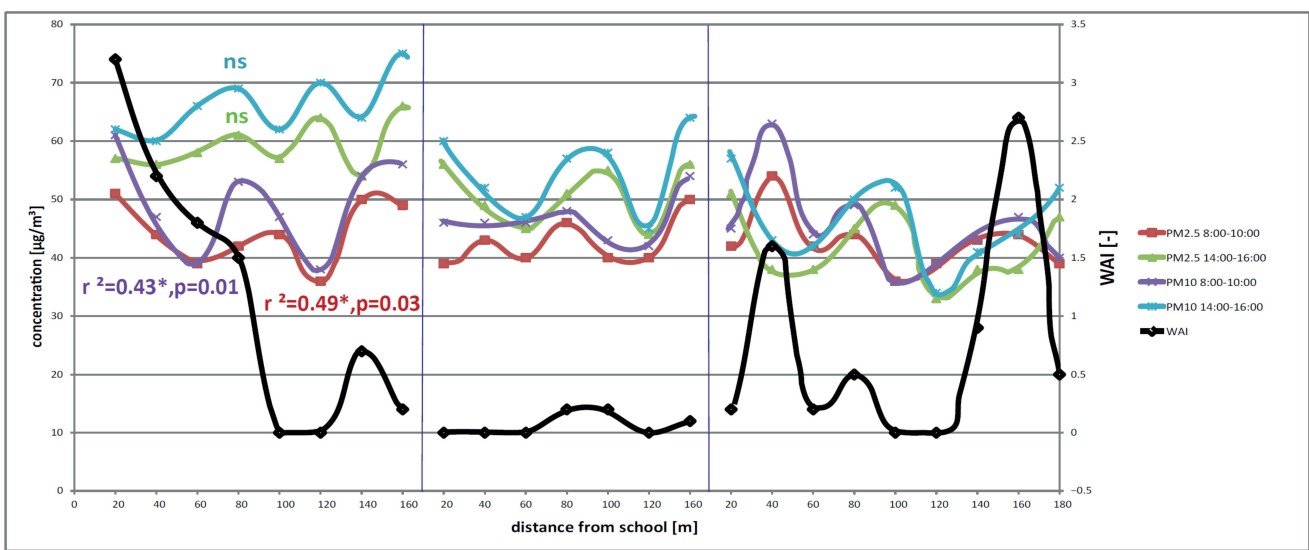

**Figure 4.** WAI distribution and an average PM concentration from 8:00 to 10:00 and from 14:00 to 16:00 in study plots along walking paths to school A (low share of trees, low number of local heating emission sources). ns—no significance.

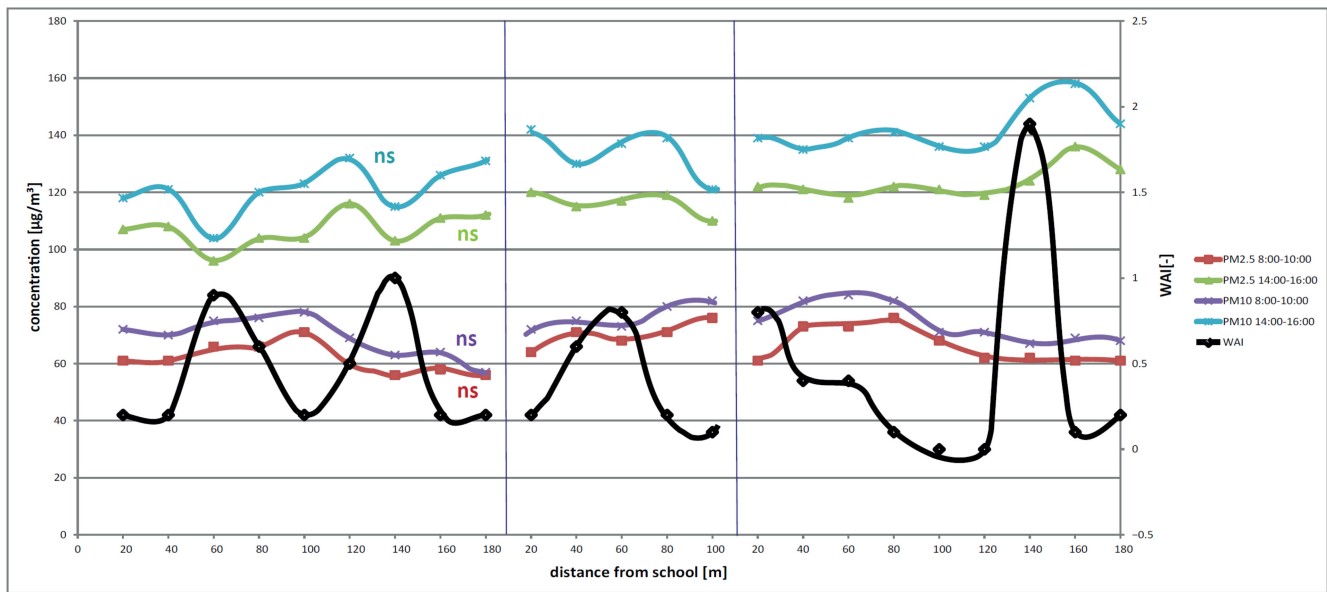

**Figure 5.** WAI distribution and an average PM concentration from 8:00 to 10:00 and from 14:00 to 16:00 in study plots along walking paths to school B (high share of trees, high number of local heating emission sources). ns—no significance.

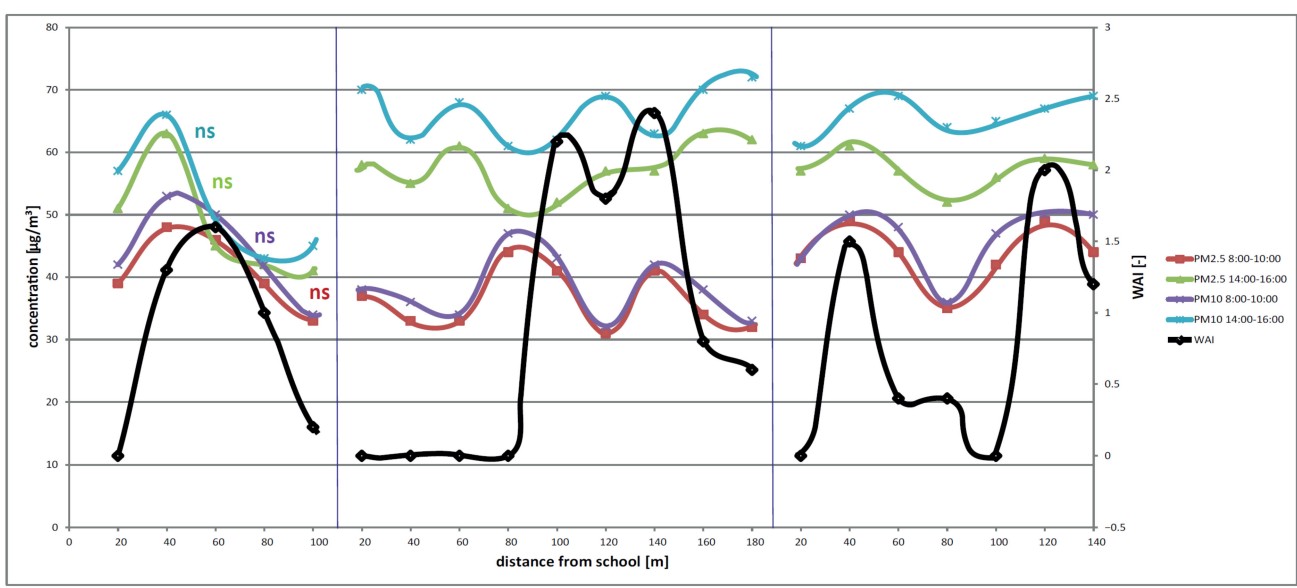

**Figure 6.** WAI distribution and an average PM concentration from 8:00 to 10:00 and from 14:00 to 16:00 in study plots along walking paths to school C (very high share of trees, low number of local heating emission sources). ns—no significance.

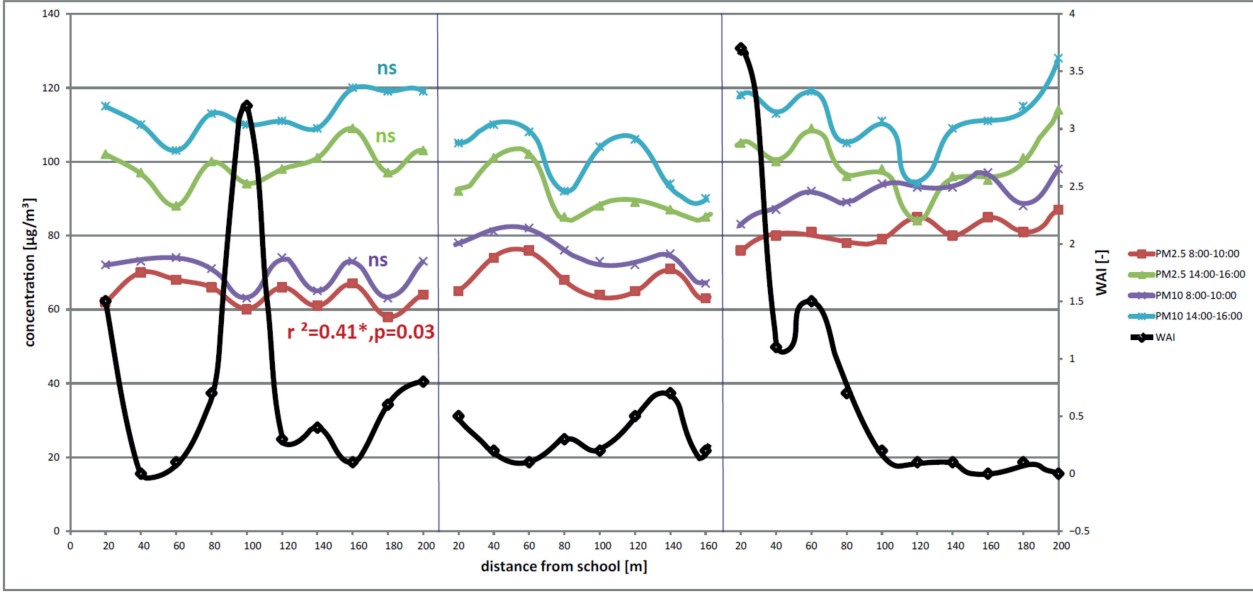

**Figure 7.** WAI distribution and an average PM concentration from 8:00 to 10:00 and from 14:00 to 16:00 in study plots along walking paths to school D (low share of trees, high number of local heating emission sources). ns—no significance.

## 4. Discussion

In Warsaw, similarly to many other European cities, where energy coal-based heating systems are still common and the transformation towards clean energy sources is still ineffective, the majority of PM is produced in winter and it is related to heating activities [41,42]. Undoubtedly, drawing out from fossil fuels in the residential sector is essential for reducing PM pollution and thereby improving the city residents health status [41]. Fundamentally improving air quality requires deep decarbonisation of the energy system, as well as more synergistic pathways to simultaneously address air pollution and global climate change [43]. However, the lack of local solutions related to CHP emissions [42]



results in very localised problems. A study of routes to schools in Warsaw found that proximity to local CHP emitters had the strongest impact on pollution on the way to schools (Figure 2). Given that the children spend more than 50% of their active time commuting to school [18], exposure to pollutants can have a critical effect on their health, which can impede pulmonary function development [1], leading to asthma development [2] and susceptibility to otitis media [3].

An example of possible actions aimed at mitigating high loads of PM is the introduction of more greenery into cities. Models on the reduction of PM dispersion by trees in cities show promising results [44], indicating that the size, distribution and species composition of vegetation play a key role in PM reduction [45]. Urban trees contribute to improving air quality and can be used in national air protection strategies to reduce air pollutant concentrations [46]. The ability to effectively capture PM is an important factor in selecting optimal plant species to be used in urban greening [47]. Particularly promising might be the evergreen plants, which keep their leaves throughout the seasons. This means that they can be used for air purification in the winter season [48]. However, current studies have not proved their efficiency in PM removal.

The presence of greenery close to schools is mostly associated with their beneficial aesthetical as well as educational role for the young generation as even the sole visibility of vegetation outside the window was linked to improved performance of school pupils, not to mention the educational and aesthetic value [19,49].

Models predicting efficiency of greening interventions have been shown to poorly capture the seasonal variability of greening, even when the main source PM is an increased residential heating in winter [50]. There is a lack of research on the role of trees in the leafless state in reducing high loads of PM, which impedes our understanding of its impact on human health and well-being. Additionally, effectiveness of vegetation and application of phytoremediation methods is mostly criticized to be effective only during the vegetation season, while some cases are reported where dense vegetation in the winter can lead to creating local concentrations of PM [51,52]. Moreover, the trees can reduce the air flow and in some cases contribute to the local increase of pollutants, due to changes in their dispersal mode [46]. Our results suggest that such situations are possible during the winter (Figures 4 and 7). However, the influence of tree density in the leafless state expressed by WAI on the formation of local pollutant concentrations is questionable (Figures 4 and 7). The investigated walking routes in places with the highest density of trees tended to stimulate PM concentrations. On the other hand, the tree stands of comparable density did not show such peaks (Figures 4 and 7). Trees, characterized by a high foliage density, can act as a barrier preventing pollutant dispersal and be used as a biological filter [25,53]. The number of locations studied does not allow us to unambiguously resolve whether WAI is significantly related to the local accumulation of pollutants. Studies performed during the vegetation season state tend to argue that pollution decreases with the density of trees [54]. A sufficiently large green area with a well-chosen species composition can be a viable way to improve air quality and in some cases even reduce PM pollution to acceptable levels [55]. In this work, the surroundings of schools B and C were the greenest (Figure 3), but this did not translate into significant reductions in PM concentrations measured at height and along the children's route to school (Figure 2). It seems that in order to effectively filter PM from the air in wintertime, urban greenery should be as numerous and dense as possible while maintaining porosity that guarantees air movement. In this way, two unfavourable phenomena, local PM stagnation and uncontrolled transport of pollutants to potentially clean locations, may be limited.

The role of trees during dormancy season in air purification processes requires further research [54]. Potentially, the rough surface of branches and the remaining withered leaves of some species, accompanied by a few evergreen and coniferous plants, could potentially have some positive effect on reducing PM loads which could be deposited and trapped on their surface, accompanied by the sheltering effect, allowing less pollutant to reach the paths used by the children. There are no studies, however, which could confirm that

phenomenon and assess its scale and extent. If it is confirmed that the occurrence of locally increased PM concentrations is a frequent phenomenon, further steps should be taken to counteract this, especially in places such as children's routes to school. Coniferous trees seem to be useful in this respect. These species also absorb PM during the winter months when air quality is poorer. Due to their smaller leaves, a larger wax layer on their needles and more complex shoot structures, they have a high capacity to capture pollutants from the air [56]. Species such as black pine or common yew [57], as well as climbers, could be grown along streets and be more effective in PM removal [58]. Undoubtedly greening interventions should always meet social approval, and the aesthetic function plays a primary role in species selection and the greening solution used. Therefore, plant species proved to be most efficient in PM reduction might not necessarily be those most meeting public preferences. However, the numerous studies, included those performed in Warsaw, show a growing approval for innovative greening interventions, showing that the public is willing to accept other-than-traditional forms of greenery, if they are supported by economic or ecological benefits [59,60].

The results we obtained did not show a positive role of trees in air purification in any of the locations. This could have been caused by the difficult environmental conditions along the roads, the discontinuity of greenery, and the relatively small proportion of trees in the area (Table 1). However, the results showing the negative impact of tree canopies during the winter season should not discourage the use of greenery, as the benefits outweigh the effects of exposure [61]. We hope to encourage further research and search for solutions in identifying these negative positions.

We suggest that areas next to schools and roads to schools should be considered to require special attention, as all locations studied were exposed to above-normal concentrations of particulate air pollution (Figure 2). Areas along school roads should be greened continuously [54] to allow ventilation. Species selection and planting structure should also take into account the most difficult winter heating period in Northern Europe, especially near local pollutant emitters.

## 5. Conclusions

- Children attending schools located near thermal heating emitters were twice as exposed to high PM concentrations.
- Tree planting on the way to school in the winter season showed no positive effect on air quality along the analysed sections of the road to the school, and locally even increased PM concentrations.
- Research on how to avoid local concentrations of pollutants should be expanded by modifying the structure of greenery and the share of evergreen species to increase the effectiveness of plants in the winter season.
- We propose to treat roads to schools as special zones of street greenery.

**Author Contributions:** Conceptualization, D.S. and A.H.; methodology, D.S., P.S. and A.H.; software, P.S.; validation, D.S., P.S. and A.H.; formal analysis, A.H., D.S., P.S. and A.P.; investigation, A.H.; resources, P.S.; data curation, D.S., P.S. and A.H.; writing—original draft preparation, A.H.; writing— review and editing, P.S., D.S. and A.P.; visualization, M.M.; supervision, P.S., D.S. and A.P.; project administration, D.S. and A.H.; funding acquisition, D.S. All authors have read and agreed to the published version of the manuscript.

**Funding:** This research was funded by the National Science Centre (Poland) grant no. 2020/39/B/ HS4/03240.

**Informed Consent Statement:** Not applicable.

**Data Availability Statement:** Not applicable.

**Conflicts of Interest:** The authors declare no conflict of interest. The funders had no role in the design of the study; in the collection, analyses, or interpretation of data; in the writing of the manuscript, or in the decision to publish the results.

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
