# Peer review of "The Role of Trees in Winter Air Purification on Children’s Routes to School"

_forests, doi:10.3390/f13010040_

Round 1

Reviewer 1 Report

The article is valuable and, as corrected. In my opinion, it just requires a broader context.

Detailed comments below: 

L 104 -105 Definition is needed - what does low tree canopy cover mean? Criteria for when is high when low tree canopy cover for urban areas should be clarified

I 230-231 think the evergreen tree theme needs to be developed - maybe this is the next step - how do evergreen species improve this balance? what do the studies say about this?

L 246-247 Tree plantings can be used to block wind speed or increase it - potentially increasing it potentially increasing - by ventilating More citations is needed

Author Response

All responses are added in the word file 

Reviewer 2 Report

In the papier enitiled: "The role of trees in winter air purification on children’s routes to school" the authors raise the topic of air pollution near ground schools, and thus the exposure of children to pollutants emitted, in particular, from black-smoke-belching stoves. The authors also set out to investigate how neighboring trees can reduce exposure to high concentrations of PM. Measurements were carried out in 4 localizations in the city of Warsaw (Poland) on 3 measurement days in the winter (leafless) in the range of PM concentration. Additionally, the number of stoves in 1km buffer and share of forested area in 500m buffer was estimated on the basis of the BDOT database. The discussed topic is interesting and the problem is important. However, the text is missing some important information that should be supplemented:

  1. line 116 - What does it mean "We we took regular measurements in plots located every 20 m, where we took measurements of PM concentrations and inventoried vegetation, discarding those sites at the edge of the tree canopy." The distance between successive measurement points was 20m? Vegetation coverage was determined for each buffer with a diameter of 20m? How? By number of trees?
  2. line 124 "on three windless days (...) at weekly intervals". Were there three repetitions of measurements in a given location (on a given route) corresponding to three measurement days? 
  3. line s 133-135 reformulate the sentences "A series of measurements from 60 seconds were recorded three times during the measurement. A reading was taken every 10 seconds, then the average was recorded." because it is not clear what was the measurement interval of 10 or 60s?
  4. lines 136-138 - please specify the way of taking measurements. The device SS1-COM-R4 measures LAI at a distance of 1 m. Cave measurements were made for a buffer with a diameter of 20 m?  
  5. line 147 - was the compliance with the normal distribution really checked? There is no information in the text on this subject - Fig.2. and Fig.3. question this thesis.

Other comments:

  1. Fig.2. and Fig.3. presents the same data. I recomend delete Fig. 3.
  2. Fig 5. - Fig. 8. measurement points cannot be connected with a line because measurements with number 3.5 or 10.3 cannot be considered. As a scientific publication, it should be factually correct. There is no basis for interpolating values between measurements.
  3. Fig 5. - Fig. 8. presents the r2 values. What are these Pearson correlations between WAI and PM concentrations?

Author Response

All responses are included in the word file

Round 2

Reviewer 1 Report

I dont have any additional comments 

Reviewer 2 Report

Thank you for reading my comments and for including them in the revised version of the work.
The normality of the distributions presented in the response is strongly debatable. They are all clearly bimodal, and it is worth considering why.
Nevertheless, after supplements, I recommend the papier for publication in the submitted version.